# A genetic toolkit and gene switches to limit *Mycoplasma* growth for biosafety applications

Alicia Broto[1], Erika Gaspari[2,3], Samuel Miravet-Verde [4], Vitor A. P. Martins dos Santos [2,5] &
Mark Isalan [1✉]

*Mycoplasmas* have exceptionally streamlined genomes and are strongly adapted to their many
hosts, which provide them with essential nutrients. Owing to their relative genomic simplicity,
*Mycoplasmas* have been used to develop chassis for biotechnological applications. However,
the dearth of robust and precise toolkits for genomic manipulation and tight regulation has
hindered any substantial advance. Herein we describe the construction of a robust genetic
toolkit for *M. pneumoniae*, and its successful deployment to engineer synthetic gene switches
that control and limit *Mycoplasma* growth, for biosafety containment applications. We found
these synthetic gene circuits to be stable and robust in the long-term, in the context of a
minimal cell. With this work, we lay a foundation to develop viable and robust biosafety
systems to exploit a synthetic *Mycoplasma* chassis for live attenuated vectors for therapeutic
applications.

[1] Department of Life Sciences, Imperial College London, London SW7 2AZ, UK. [2] Laboratory of Systems and Synthetic Biology, Wageningen University &
Research, Wageningen, the Netherlands. [3] European & Developing Countries Clinical Trials Partnership (EDCTP), The Hague, The Netherlands. [4] Centre for
Genomic Regulation (CRG), The Barcelona Institute of Science and Technology, Dr Aiguader 88, 08003 Barcelona, Spain. [5] LifeGlimmer GmbH,
Berlin, Germany. ✉email: m.isalan@imperial.ac.uk

*M*ycoplasmas are perhaps best known for their reduced genome sizes, making them appealing minimal-cell models. When whole-genome sequencing began in the 1990s, *Mycoplasma genitalium* and *Mycoplasma pneumoniae* were among the first bacteria to be sequenced[1,2]. Since then, *Mycoplasmas* have been intensively exploited in systems biology studies and whole-cell modelling projects[3]. Moreover, they have been at the centre of significant milestones in synthetic biology, including the genome transplantation technique[4], the first chemically-synthesised bacterial genome[5], and the first synthetic cell[5]; achievements that culminated in the design of a minimal bacterial genome[6].

Due to their genomic austerity, *Mycoplasmas* exhibit limited biosynthetic capabilities. Their strict nutritional needs result in a parasitic lifestyle for most *Mycoplasma* species[7], which has made them a major problem in animal husbandry. For instance, every year infections caused by *Mycoplasmas* in poultry, cows and pigs produce multimillion-euro losses worldwide. These bacteria lack cell walls and so are resistant to many of the common antibiotics[8–10]. Furthermore, there are no vaccinations available against many *Mycoplasma* species infecting humans and animals (e.g. *M. bovis*[11]). In fact, there are only three effective vaccines on the market, all of them based on large-scale production of attenuated or inactivated *Mycoplasma*[12–14]. Increasing the repertoire of such vaccines is, therefore, desirable.

Recently, efforts have been focused on engineering a *Mycoplasma*-based cell chassis for vaccines and other applications. One of the most advanced projects[15] is capitalising upon the omics data generated for *M. pneumoniae*[2,16–32]. This is being used to design a universal *Mycoplasma* chassis that can be deployed as a single- or multi-species vaccine in a range of animal hosts, by displaying selected surface antigens. The main idea is to use genome engineering of *M. pneumoniae* to remove pathogenic determinants and so develop a non-pathogenic chassis, which will also retain some interesting *Mycoplasma* features, such as the lack of cell walls, limited metabolic capabilities and a special genetic code[33–35]. Such a cell chassis will have the potential for synthetic biology applications[36] (e.g. drug delivery systems[37]) as well as providing a scaffold for vaccines. However, developing an *M. pneumoniae* chassis as a live vector will require the implementation of biosafety measures to maximise containment of the resulting strain. In this regard, synthetic gene switches or kill-switches[38] can provide genetic barriers limiting *Mycoplasma* growth to bioreactors during bioproduction. One of the main aims of the present work is therefore to develop such switches using transcription factors.

*M. pneumoniae* presents a very limited set of endogenous transcription factors: only nine in total (compared with around 400 in *Escherichia coli*), including only two sigma factors[23]. The study of *Mycoplasma* biology is also limited, given the reduced amount of molecular genetic tools available. For example, while inducible gene expression systems have been widely exploited in model bacteria, like *E. coli* or *Bacillus subtilis*, that is not the case for *Mycoplasma*, despite their huge potential as cell chassis. With new advances in genome engineering techniques[39–43], it is more important than ever to increase the gene regulation toolkit for these microorganisms.

In terms of developing gene regulation tools for *Mycoplasma*, a decade ago the tetracycline-inducible promoter Pxyl/tetO₂ from *B. subtilis* was shown to be functional in the animal pathogen *Mycoplasma agalactiae*[44]. Since that first report, this tetracycline-inducible promoter has only been exploited in two other *Mycoplasma* species: *M. genitalium*[45] and *M. pneumoniae*[39]. Nonetheless, the use of this system is still rather limited, as it is the only inducible promoter available. Increasing the amount and versatility of such tools, and porting other well-used tools from other model bacteria,

will contribute to the exploitation of *Mycoplasma* as a synthetic biology asset.

In this study, we present the adaptation to *M. pneumoniae* of three well-characterised inducible systems: AraR from *Bacillus subtilis*[46], LacI systems from *E. coli*[47,48] and the CI system from phage lambda[49,50]. We then expanded the versatility of the TetR system from *E. coli*, and tested synthetic theophylline-riboswitches in *M. pneumoniae*. With this genetic toolkit at hand, we subsequently engineered synthetic gene switches to control and limit *Mycoplasma* growth for biosafety containment applications. Finally, we ascertained the long-term stability and robustness of these synthetic gene circuits in the context of a minimal cell. With this work, we provide the initial steps to developing viable biosafety systems to help exploit a synthetic *Mycoplasma* chassis for live attenuated vaccines[51–53] or even for a live biotherapeutics (LBT) vector[54–58].

## Results

**A gene regulation toolkit for *Mycoplasma*.** A choice of orthogonal gene expression systems is currently missing from the genetic toolkit available for *M. pneumoniae*. Until now, only one tetracycline-inducible promoter from *B. subtilis* has been reported to function in *Mycoplasma*. To overcome this limitation, we set out to engineer three well-known repressor systems to be orthogonal genetic switches in *M. pneumoniae*: the AraR system from *B. subtilis*[46], the LacI system from *E. coli*[47,48] and the CI system from phage lambda[49,50].

First, we set out to establish the expression of the transcription factors (TFs) AraR, LacI and CI in *M. pneumoniae*. Genes coding for the repressor AraR and the L-arabinose permease AraE were isolated from genomic DNA from *B. subtilis* and expressed suitably (Supplementary Fig. 1), although later we showed that AraE is not needed for AraR to function in *M. pneumoniae* (Supplementary Fig. 3). By contrast, we used codon-optimised genes for the expression of LacI and CI proteins (Supplementary Fig. 1). The constitutive promoter p438 from *M. genitalium*[59] was successfully used for the expression of all these proteins, except for LacI. To reach a suitable expression level of the LacI protein, it was necessary to use a stronger synthetic promoter (pS) and to engineer the 5′-end of the gene to insert the coding sequence for a short leader-peptide to improve translation and stability of the protein in *M. pneumoniae*. This leader aims to reduce the secondary structure of the 5′-end of the gene, as reduced secondary structure correlates with increased expression in bacteria[60] and this correlation may be even more important in a bacterium that does not depend on RBSs for translation initiation[61]. As shown in Supplementary Fig. 1, all proteins were thus successfully expressed in *M. pneumoniae*.

The next step was to engineer promoters with operator sites for the TFs (AraR, LacI and CI). Additionally, we designed promoters with TetR-operator boxes to gain the versatility of the tetracycline-inducible system for *M. pneumoniae*. We used the promoter consensus sequence of *M. pneumoniae*[32] in the design of promoter/operator pairs. Moreover, we took into account each TF-operator consensus sequence, the amount and position of the operator boxes inside the promoter region, the structure of the repressor binding to the DNA and the influence of DNA-bending[46,48–50,62–73]. The promoter candidates that we designed were first evaluated in silico using the same algorithm developed to predict promoters in the genome of *M. pneumoniae*[32]. We also included in the analysis a few heterologous inducible promoters extracted from the original hosts of the systems, but most of them did not meet the threshold of the test (Supplementary Data 1A). Subsequently, candidates predicted to be a promoter sequence in

*M. pneumoniae* were synthesised and cloned into a minitransposon (MTn) platform. The latter includes a transcriptional terminator to insulate the promoter activity and a gene encoding an *mCherry* reporter (Supplementary Fig. 2A). These MTns were used to generate polyclonal *M. pneumoniae* mutant strains. Thus, we analysed the activity of the synthetic promoters in a mCherry time-course assay (Supplementary Fig. 2). To determine the background noise in these experiments, we used a strain carrying an empty MTn. Also, we used a strain with an MTn with the constitutive promoter p438 as a positive control for the mCherry signal. All synthetic promoters tested produced mCherry expression in *M. pneumoniae*. Additionally, we confirmed mCherry expression in the late-exponential phase of growth by western blot (Supplementary Fig. 2B). Importantly, we achieved a wide range in the strength of the synthetic promoters (Supplementary Fig. 2). Also, the results confirmed that all predictions made in silico turned out to be functional promoters in *M. pneumoniae*.

To study the complete inducible systems in *M. pneumoniae*, we generated polyclonal strains by re-transforming the strains carrying the TFs with the different MTn expressing mCherry via the synthetic promoters. In this case, we performed the mCherry time-course assays at different concentrations of the corresponding inducer. Similar to the previous time-course assays, we used a strain without the *mCherry* gene as a negative control and a strain with the *mCherry* gene with the constitutive promoter p438 as a positive control. In the positive control, as expected, the expression of mCherry is independent of any of the TFs or inducers (Supplementary Fig. 4). By contrast, strains expressing the relevant TF alone produce little mCherry signal with the inducible promoters, suggesting a limited basal level, which is a desirable feature. The addition of the inducer to the medium reveals a clear mCherry signal, and that signal increases with increasing amounts of inducer. We confirmed the variations in mCherry expression depending on the inducer concentration in the late-exponential phase by western blot or fluorimetry (Fig. 1). In this way, we showed that all inducible systems are functional in *M. pneumoniae*. Induction of the reporter is dose-dependent in *M. pneumoniae* and all systems appear to work with promoters with very different strengths (Fig. 1).

To complete the toolkit, we evaluated several synthetic theophylline-dependent riboswitches[74,75] in *M. pneumoniae*, using the MTn-reporter platform. The results show that theophylline-dependent riboswitches are functional in *M. pneumoniae* but with relatively high basal levels and narrow dynamic ranges (Supplementary Fig. 5). Recently, a transcriptional study in *M. pneumoniae* used a high-throughput screening system based on ELM-seq methodology[21]. It would be interesting to apply this methodology to find better riboswitches for *Mycoplasma* in the future.

**Engineering synthetic gene circuits in *Mycoplasma*.** We applied the tools developed above in the design of gene switches to limit *Mycoplasma* growth, for example for live vaccine chassis safety switches. The first case studied was a kill-switch designed to generate fatal toxicity by producing multiple double-strand breaks (DSB) on the *Mycoplasma* chromosome. Such damage is produced by a nuclease and, as *M. pneumoniae* cells lack an efficient system to repair DSB, the cells consequently die. The design of a kill-switch is usually a challenging case for circuit engineering, because of the need for absolute ON and OFF states, but it is even more difficult when working with an organism that has a promiscuous transcription and translation systems, which generate a large amount of noise[19,23,*]. We, therefore, tested several kill-switches and nucleases. Notably, the toxicity burden imposed on the system by the slightest basal expression of a toxin, a nuclease, in this case, was the main limitation for a successful

kill-switch in *Mycoplasma* (see Supplementary Fig. 6 for examples). We only overcame this problem when using an RNA-dependent site-targeted nuclease, the Cas9 protein from *Streptococcus pyogenes*[76]. In this case, the need of two elements (the nuclease and the RNA) to generate a DSB increases the toxicity threshold, and the system tolerates a more-achievable basal expression of the toxin. Nonetheless, for a successful kill-switch, we still had to optimise the induction of Cas9. For that, we engineered a weak inducible promoter (pLG64) and, more importantly, we changed the initiation codon from ATG to a weak CTG start codon. Thus, we managed to reduce basal Cas9 expression to below detectable levels by western blot while still inducing enough Cas9 to produce toxicity in the presence of a guide-RNA (gRNA)[77] (Fig. 2). This kill-switch (KS1) is an IPTG switch for Cas9 expression, combined with the constitutive expression of a gRNA that targets a repeated-sequence in the chromosome (Fig. 2). Polyclonal strains with the KS1 inserted in the genome by transposition have similar growth with or without gRNA in the construct, indicating a good silencing of *cas9* before induction. Conversely, severe growth impairment (reflected in an elongated lag-phase) is only observed in the strain carrying a gRNA upon the addition of the inducer IPTG in the medium, indicating some successful induction of cell death. After this initial phase, escape mutants appear to restore growth and after the first round of IPTG treatment, the population becomes insensitive to further IPTG-induced killing (cf. later experiments below, with sequence analyses of escape mutants and the eventual implementation of designs to obviate these effects). The circuit with multiple single-target gRNAs that promote DSB in different genes (instead of a single gRNA with multiple targets) has similar performance (Supplementary Fig. 7). We also re-engineered the kill-switch in a way that the LacI represses the *cas9* and two gRNAs simultaneously (Supplementary Fig. 7). Again, this design is fully inducible but performs similarly to the original KS1. Therefore, the silencing and induction of a single *cas9* gene appears sufficient to reduce growth, but only for a certain period of time, and so we decided to search for additional kill-switch designs to complement KS1.

We engineered a second kill-switch (TKS2) using the same nuclease, Cas9. This time, we focused on engineering a synthetic addiction that would maximise the containment of *M. pneumoniae* when combined with a kill-switch. For that, theophylline (Theo, a methylxanthine chemically-related to caffeine) is a good candidate since it is effectively absent in animal tissue in vivo[78]. Thus, a theophylline-dependent switch could prevent unwanted growth in a vaccinated host. The TKS2 was designed as a positive feedback loop that incorporates a theophylline-riboswitch to regulate the expression of the main TF in the circuit, the TetR (Fig. 3; for intermediate optimisation steps see Supplementary Figs. 9, 10). The inhibition of TetR with tetracycline, or its analogue anhydrotetracycline (ATc), activates the circuit. Similarly, the lack of theophylline is designed to achieve similar results by decreasing TetR expression (Fig. 3). Either way, the loss of TetR triggers the expression of the nuclease Cas9, the Cre recombinase and the CI repressor, with all three elements arranged in a single operon (Fig. 3). The Cre recombinase in the circuit acts here as an indirect repressor of the TetR. To achieve that, we included a pair of lox sites flanking the gene encoding the TetR repressor (*tetR*). Hence, the induction of Cre eliminates TetR from the system by recombining the pair of lox sites. Moreover, we made the activation of the circuit irreversible with a pair of mutant lox sites (lox66 and lox71). Finally, the TKS2 also includes a CI for its potential to be exploited as a repressor of a *Mycoplasma* conditional-survival gene. Although we successfully engineered some *Mycoplasma* promoters to include CI-operators (Supplementary Fig. 8), we found that these did not improve

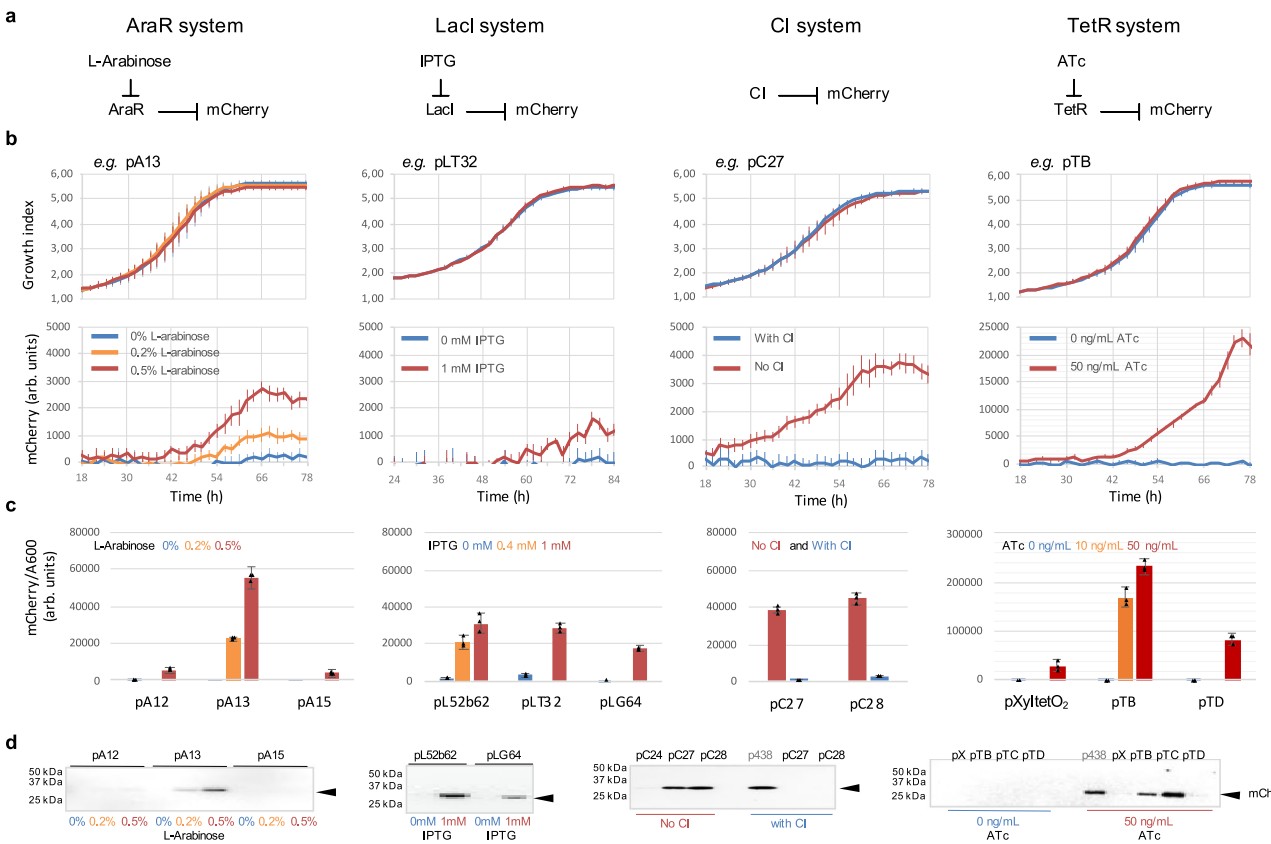

**Fig. 1 First genetic toolkit for *Mycoplasma*. a** Schematic representation of the inducible systems based on the AraR repressor (L-arabinose inducer), LacI repressor (IPTG inducer), CI repressor, TetR repressor (ATc inducer). Blunt arrows indicate repression. **b** Induction time-course results for one synthetic promoter selected for each repressor system. The upper graphs show synchronised growth (growth index as the ratio Abs430 nm/Abs560 nm) between uninduced (blue) or induced (red and orange) cultures with the corresponding inducer for each repressor. The lower graphs show the corresponding mCherry kinetics (fluorescence in arbitrary units; background subtracted for autofluorescence—see Methods). The mCherry expression depending on the inducer concentration was confirmed by Fluorimetry (**c**) and western blot (**d**) in the exponential phase (two days after induction) for several synthetic promoters in each repressor system. All results from mean values from three bio-replicates (black triangles show the values of each bio-replicates). Error bars indicate standard deviation. Source data are provided as a Source Data file.

circuit function (Supplementary Fig. 10) and so these were not included in the final TKS2 design. For now, the *cI* gene has a passive role in the stabilisation of the circuit before induction (Fig. 3b).

We first optimised the circuit TKS2 in the absence of a gRNA. The TetR expression, crucial to prevent leaky expression of the toxic elements, was initially insufficient even in the presence of theophylline (Fig. 3). We found we had to duplicate the riboswitch-*tetR* cassette in the circuit to improve its expression significantly to provide the desired silencing of the kill-switch (Fig. 3). As explained above, for a successful kill-switch, limiting the basal expression of the toxins is the most important thing. In this circuit, preventing the leaky expression of the Cre recombinase was even more important. Similarly to Cas9 induction, we managed to decrease Cre expression, below the western blot detection limit, by mutating the initiation codon to CTG, adding nine suboptimal codons among the twenty codons at the beginning of the *cre* gene and by avoiding putting *cre* in the first position of the polycistron (Supplementary Figs. 9, 10). We observed the induction of Cre by the reduction of the size of the PCR product generated when amplifying the fragment of the circuit flanked by the lox sites (Fig. 3b, c). We also confirmed the ATc-induction of the three elements (CI, Cas9 and Cre) by western blot or PCR (Fig. 3c, d). Moreover, the dilution of the theophylline that occurs when passaging the strain in a theophylline-free medium also produced increasing amounts of the three elements and resulted in

the programmed loss of the TetR (Fig. 3d). Therefore, the network components appeared functional.

We next inserted the circuit TKS2 by transposition into a strain that already had the KS1 circuit with a functional gRNA. The resulting double kill-switch strain induces strong growth impairments with either IPTG or ATc. The best result is achieved when both inducers are added together to trigger both circuits simultaneously (KS1 and TKS2). In that case, no growth was observed even during 10 days of incubation, indicating a strong kill-switch effect (Supplementary Fig. 11). Transferring the strain into a theophylline-free medium by itself has a small impact on growth. Nevertheless, this is to be expected as in this case only one of the two kill-switches in the strain would be activated. Overall, the double circuit functions as a strong and efficient *Mycoplasma* kill-switch, in the presence of the two small-molecule inducers.

**Long-term kill-switch circuit stability and robustness in the context of a minimal cell.** The optimisation and analysis of the circuits were initially studied using polyclonal strains with the circuits inserted in the genome by transposition. Later, we isolated clonal strains and determined the location of the circuits in the genome by sequencing. We grew the clonal strains by sequential passaging every 48 h for a month (15th passages) to study the long-term stability and robustness of the biosafety systems.

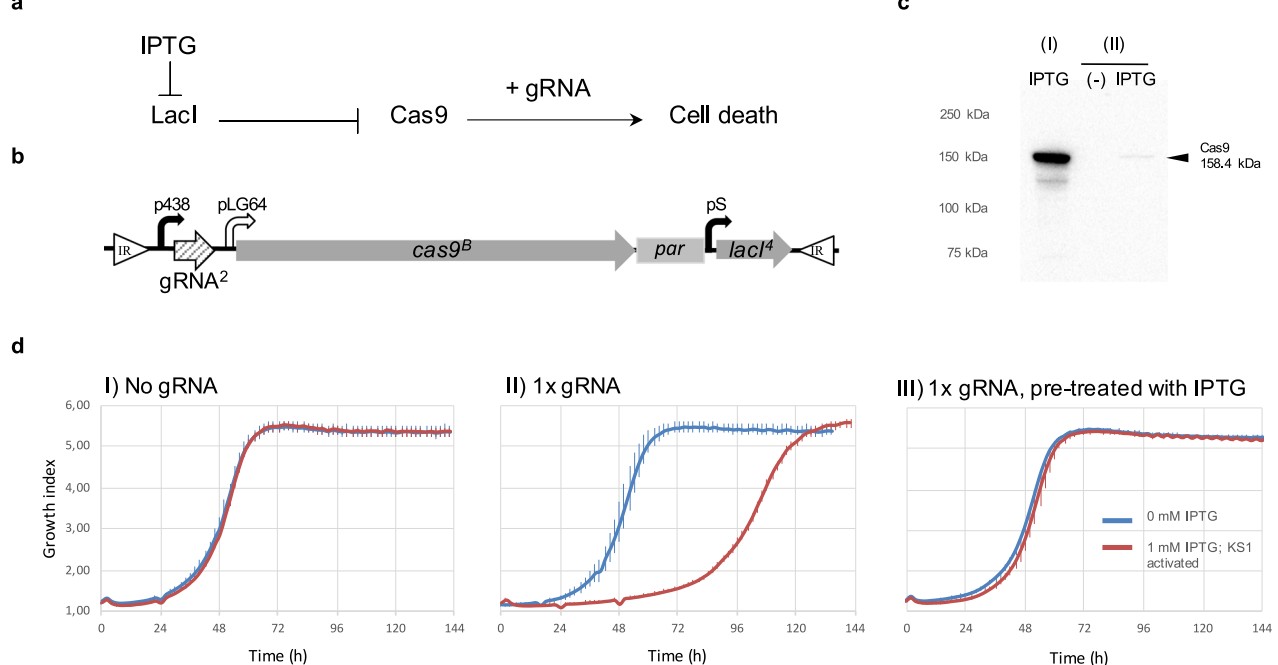

**Fig. 2 Design and characterisation of the kill-switch KS1. a** Schematic representation of the kill-switch KS1. Blunt arrows indicate repression and pointy arrow activation. **b** Schematic representation of relevant DNA segments of the MTn*par* with the KS1 used to generate a polyclonal strain (puromycin used for selection). White triangles represent the inverted repeats flanking the fragment of the MTn inserted in the genome of *Mycoplasma* (IR); Bend-arrows for constitutive promoters (black) and IPTG-inducible (white) promoters. Arrows for genes (grey), gRNA (stripped; gRNA[2] has eight targets in the genome of *M. pneumoniae*); *par*, selection cassette. **c** Western blot results showing Cas9 expression in strains with the KS1 kill-switch with the *cas9* gene with the canonical initiation codon ATG (I) or a CTG codon instead (II). Protein expression was analysed using 10 µg of total protein extract from pre-induced (−) and 1 mM IPTG-induced (IPTG) samples. **d** Growth kinetics of polyclonal strains with KS1 without any gRNA (I), with a single gRNA copy (II) and same but pre-treated with IPTG in the previous passage (III). Graphs show growth kinetics (growth index corresponding to the ratio Abs430nm/Abs560nm) when KS1 is uninduced (0 mM IPTG, blue lines) or IPTG-induced (kill-switch activated with 1 mM IPTG, red lines). Mean values from three bio-replicates. Error bars indicate standard deviation. Source data are provided as a Source Data file.

As explained above, a polyclonal strain with the single biosafety circuit KS1 has severe growth impairments in the IPTG-medium, but killing is not complete as some individuals within the strain restore growth and so the population becomes immune to the kill-switch. We analysed this further with the clonal strain C5, which has a single minitransposon with the single KS1 kill-switch inserted into the *mpn462* gene (encoding a hypothetical protein), classified as non-essential in *M. pneumoniae*[17]. As expected, the IPTG-induced killing produces a much stronger growth impairment in the C5 strain compared to the polyclonal strain (Fig. 4, Supplementary Fig. 12, and Supplementary File 2). Nonetheless, the escape frequency (E.F.) of this strain, calculated as the ratio between the IPTG-insensitive CFUs and the total CFUs obtained in rich medium, is initially $7 \times 10^{-6}$ but increases rapidly with repeated growth, becoming $2.5 \times 10^{-4}$ at the 15th passage.

When sequencing the circuits of some IPTG-insensitive clones, we found *cas9* frameshift mutations that explain the inactivation of this single integrated circuit. Mycoplasmas have high mutation rates[9,79] and, the accumulation of mutations that inactivate the circuit is expected. However, the question is how resilient is the kill-switch to mutation and whether it is more sensitive to the accumulation of mutations. Therefore, we did the whole-genome sequencing of the strain C5 at different passages (p2, p3 and p15) or after IPTG-treatment at passage 3 (p3[IPTG]). Subsequently, we performed variant calling analyses to explore potential mutations in the population that could be generating escapees for the kill-switch mechanism (Supplementary Data 2). For this, we analysed two bio-replicates for each sample (or three in the case of passage p2). We first evaluated the frequency per base at which variants

are found, considering the kill-switch, and compared to genes in *M. pneumoniae* and its intergenic regions. Although the kill-switch consistently presents higher variant frequencies (2.9% on average for samples p2, p3 and p15, and 2.4% for p3IPTG), these frequencies are only significant for the sample p2 when compared to *Mycoplasma* genes and intergenic regions (Mann–Whitney *U* one-tailed test, *p* value = 0.04; *p* value >0.05 in the other cases) (Supplementary Fig. 13). Nevertheless, taking the variant frequencies normalised by the total reads matching the reference, the kill-switch for non-induced samples presents fractions comparable to those found in genes and intergenic regions (median fraction of 1.49%, 1.55%, 1.47%, respectively). In all passages analysed, we observed that the kill-switch fractions tend to be equal or lower than in the other genomic distributions suggesting that mutations in the kill-switch are not more frequent than expected in the genome (Supplementary Fig. 13).

Focusing on the kill-switch, the analysis of the two p3[IPTG] replicates revealed a group of more-frequent variants present with a fraction equal to or higher than the population median (i.e., 1.5%) (Supplementary Data 3). These variants, specified in Table 1, affect either the *lacI*[4] or *cas9*[B] genes, including missense (*n* = 16), frameshift (*n* = 3) and stop-gaining mutations (*n* = 4) (Supplementary Fig. 14). The most frequent variant is reproducible between the two replicates. It is found in the *lacI*[4] gene and affects one of the three critical residues for IPTG binding (R208 in LacI[4], corresponding to R197 in *E. coli* LacI[47]): a stop-gaining mutation in one replicate (R208*; fraction = 25.3%), which truncates the protein but preserves the DNA-binding domain, and the other a missense mutation preventing IPTG binding[47]

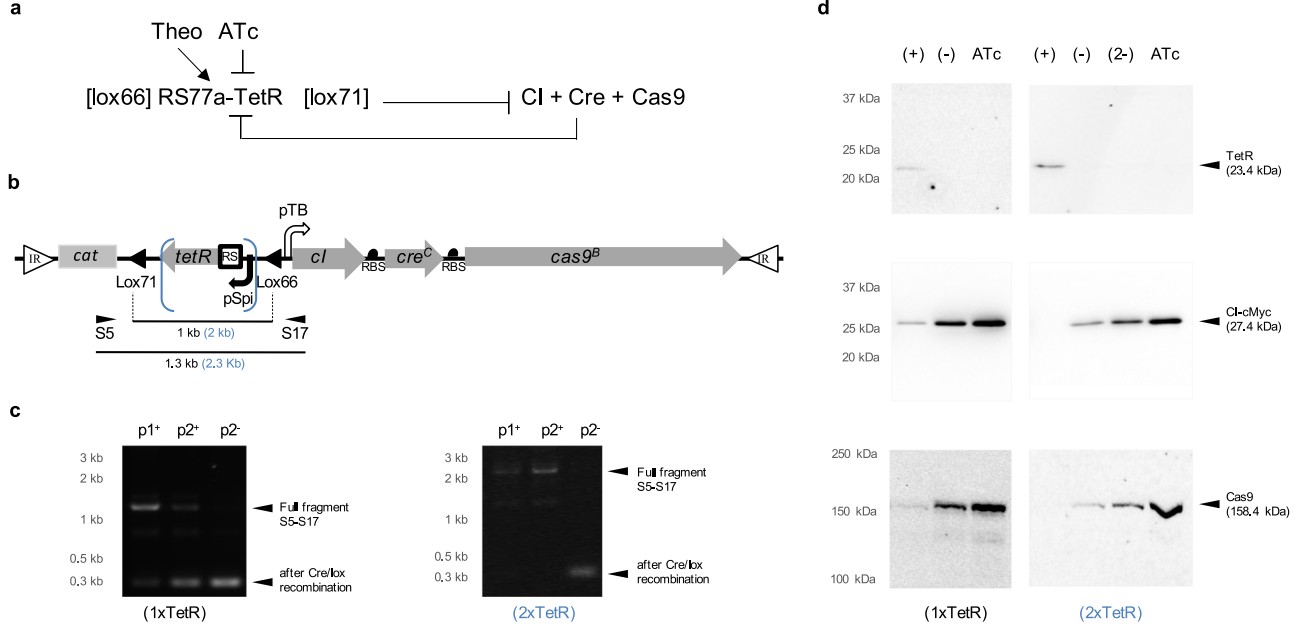

**Fig. 3 Design and characterisation of circuit TKS2. a** Schematic representation of the circuit TKS2. Blunt arrows indicate repression and pointy arrow activation. **b** Schematic representation of relevant DNA segments of the MT*ncat* with TKS2 used to generate a polyclonal strain (chloramphenicol used for selection). White triangles for the inverted repeats (IR); Bend-arrows for constitutive promoters (black) and ATc-inducible (white) promoters. Arrows for genes (grey); Black semicircles show the RBSs; A white square for the riboswitch (RS); Black triangles indicate the position and orientation of the lox sites (lox66 and lox71); *cat*, selection cassette. Blue brackets indicate the segment that is duplicated in the analysis of the circuit with 2xTetR cassette. **c** DNA gel showing the results from the PCR analysis of Cre/lox recombination in the TKS2 from genomic DNA samples from passages 1 and 2 in theophylline medium with Cre basal expression (p1+ and p2+) and passage 2 in theophylline-free medium and Cre induction (p2-). PCR product size is expected for the full fragment, before Cre/lox recombination, and the distance between lox sites are indicated in (b) between the small head arrows S5 and S17, showing the position of the primers pair used in the assay (in blue for 2xTetR). **d** Western blot results showing TetR, CI and Cas9 expression in strains with the circuit TKS2. Expression of the proteins was analysed using 10 μg of total protein extracts from incubations of the strains in the growth permissive condition of 0.5 mM theophylline (+); after a single passage in theophylline-free medium producing slow activation of the TKS2 (−); after two passages in theophylline-free medium (2−) or in 50 ng/mL of ATc to trigger the kill-switch (ATc). Results presented from the circuit engineered with one or two copies of the pSpi-riboswitch-tetR cassette between the lox pair sites in the circuit (1xTetR or 2xTerR, respectively). DNA gel and western blot results are representative of two independent experiments in each case.

(R208C; fraction = 20.9%). The following most frequent variants correspond to frameshift mutations in the *cas9B* gene truncating the protein and abolishing cleavage activity[80] and missense mutations in the *lacI4* gene affecting the hydrophobic surface of the IPTG-binding pocket[47] (Table 1).

It should also be noted that we find a wide range of scarce variants in all samples, most of them represented by a minimal number of reads and sometimes just one (Supplementary Data 2 and Supplementary Fig. 14). However, considering the reproducibility of the most frequent variants and their expected impact in abolishing the killing activity of the circuit, they alone could explain the population that survived and became insensitive to the IPTG stress.

Taken together, these results indicate that we managed to silence the basal levels of the kill-switch very effectively and, consequently, it is not producing a significant selection pressure in the strain without chemical induction, which is a remarkable achievement. After induction, we still find a limited group of variants that affect crucial elements of the kill-switch mechanism and together explain the gain of resistance to the kill-switch. Given that the accumulation of mutations is an intrinsic property of the strain, the only way to increase the resilience of the kill-switch is to introduce redundancy in the system, as follows.

Effective redundancy in the system was first attempted by including multiple copies of the whole circuit KS1. The clone C20, derived from the parental strain C5, has a second minitransposon with the circuit KS1[3g] inserted in the intergenic region between

*mpn648* and *mpn649* genes. This strain, with two copies of the kill-switch, shows no growth during a period of 10 days when treated with IPTG, indicating a strong kill-switch effect (Fig. 4 and Supplementary Fig. 12). The escape frequency for this strain is below the detection limit $(3 \times 10^{-9})$, as no colonies were found in plates with IPTG. We also performed a variant calling analysis from the whole-genome sequencing results of the strain C20, at early passages p2 and p3, and the late passage p15. For C20 samples, the frequency of variants per base decreased compared to C5 (0.9% on average), remaining significantly lower when compared to genes and intergenic regions (Mann–Whitney $U$ one-tailed test, $p$ value <0.05 in every comparison). We evaluated whether this lower frequency could be an effect of the shared sequences between the two copies of the kill-switch (KS1 and KS1[3g]) but the total coverage per base within the kill-switches for C20 samples was in fact higher than in C5 samples (Supplementary Fig. 13). As in the previous case C5, in the absence of any pressure, mutations appear to be accumulated evenly across the whole chromosome and so the chances of inactivating the kill-switch are fairly low. In this case, it was not possible to sequence samples after IPTG-stress because no sample could be produced with this very strong kill-switch. By applying regression models to the parameters of the growth curves, we estimated the arise of escape mutants over strain passages, so to predict the number of passages that *M. pneumoniae* would take to inactivate the kill-switches in the strains C5 and C20. The simulation shows that the clone C5 is predicted to inactivate the circuit after 20 strain passages, whereas clone C20 does so after

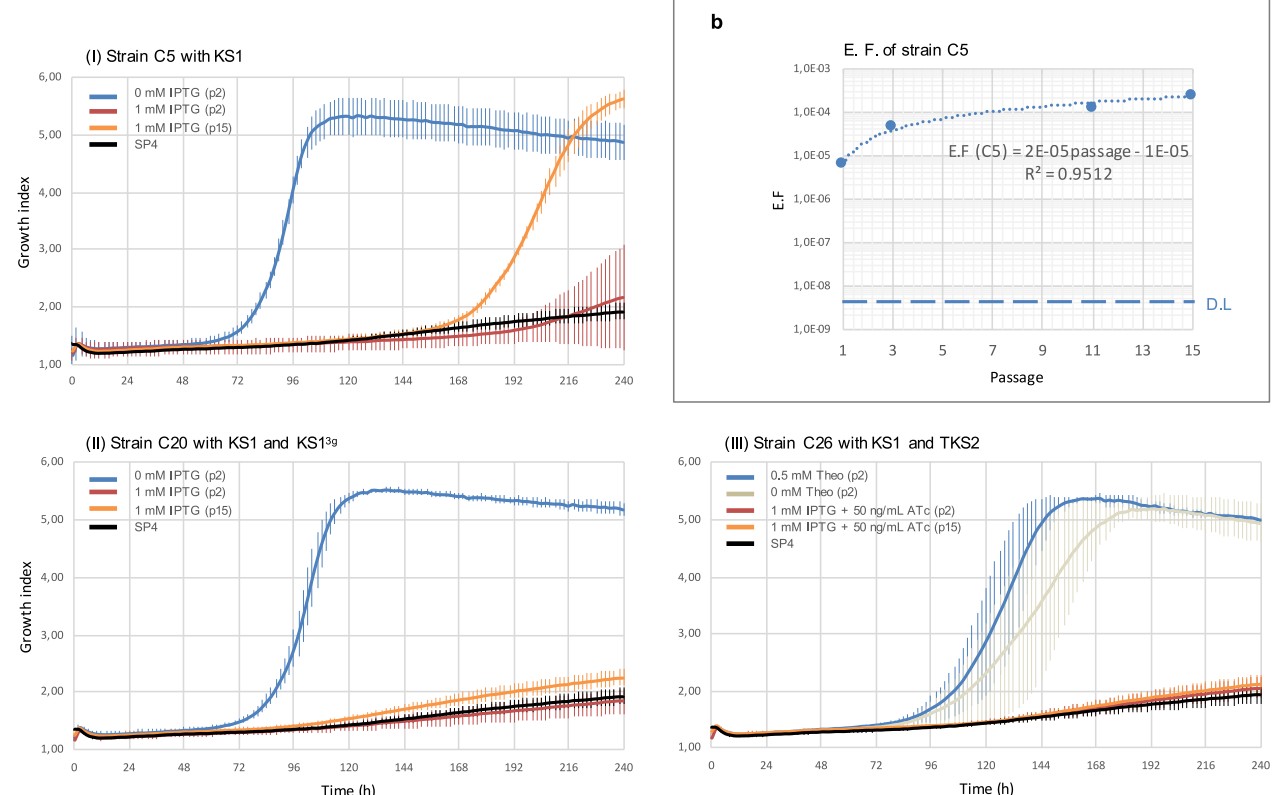

**Fig. 4 Study of the long-term performance of the three biosafety systems in clonal *M. pneumoniae* strains. a** Growth kinetics (growth index corresponding to the ratio Abs430 nm/Abs560 nm) of the clonal strains C5 (I), C20 (II) and C26 (III) analysed at an early (p2) and late passage (p15) at different induction conditions. We compare the growth of strains C5 and C20 (with one and two copies of the kill-switch KS1, respectively) when uninduced at the early passage p2 (0 mM IPTG, blue lines) or IPTG-induced that triggers the kill-switch at the passage 2 (1 mM IPTG; p2, red lines) or passage 15 (1 mM IPTG; p15, orange line). We analysed the growth of strain C26 (with KS1 and TKS2) in theophylline-medium (0.5 mM Theophylline, permissive growth conditions with both circuits inactive, blue line), or theophylline-free medium (0 mM Theophylline, slow activation of TKS2, light brown line) in the early passage p2. The growth is also analysed when both circuits are activated simultaneously in ATc and IPTG medium at passage 2 (1 mM IPTG and 50 ng/mL ATc; p2, red line) or passage 15 (1 mM IPTG and 50 ng/mL ATc; p15, orange line). The black line is a negative control of the growth index reads from an empty medium and shows the baseline of these long experiments, as the Abs560 reads have a drift (SP4, black line). Mean values from three bio-replicates. Error bars indicate standard deviation. **b** Escape frequency (EF) for strain C5 at several strain passages, determined as the ratio between IPTG-resistant CFUs and the total CFUs. The dashed line shows the detection limit calculated in the analysis, considering the total CFUs plated. Source data are provided as a Source Data file.

30 strain passages (Supplementary Fig. 15). The two predicted growth curves are compared to data for passages p2, p7 and p15 of clone C5 and for passages p2 and p15 of clone C20. Such high passages could be easily avoided during industrial production of vectors by keeping low-passage stocks and resequencing single clones if needed, as part of quality control. We were also unable to insert a third copy of the KS1 in the cells, even after several attempts at transformation. The third copy likely increases the basal expression of Cas9 to a point that is too toxic for the cells. This suggests that cells cannot overcome the system with three copies of the kill-switch.

Another potentially interesting biosafety system with some inbuilt redundancy is the combination of circuits KS1 and TKS2, which includes theophylline biocontainment. We studied the long-term stability of this system with the clone C26. This clonal strain, also derived from the parental C5, has a second minitransposon with TKS2 inserted into the *mpn062* gene (purine nucleoside phosphorylase, *deoD*). This gene was classified as non-essential at least in one of the essentiality studies in *M. pneumoniae*[81]. This was confirmed by simulating in silico the disruption of each gene comprised in the most updated metabolic

model of *M. pneumoniae* (iEG158_mpn)[82], which showed that the disruption of *mpn062* would decrease growth by only 1.2% (Supplementary Data 4). The C26 clone grows slightly slower compared to strains C5 and C20, which might be due to the disruption of this *deoD* gene. As expected for a strong double kill-switch, no growth was observed during 10 days of treatment with IPTG and ATc (Fig. 4). The escape frequency for clone C26, when both circuits are activated, is below the detection limit, as no colonies were found in plates with IPTG and ATc. However, the escape frequency of this clone C26 in ATc alone is higher, suggesting that circuit TKS2 is leakier compared to KS1 and produces a small selection pressure on the strain before proper circuit activation. Whole-genome sequencing results of samples from early and late passages of C26 (passages p2 and p15 in the presence of theophylline) revealed no significant differences in the sequences of the circuits in terms of frequency of variants per base and fraction of alternative read counts, compared to C5 (Mann–Whitney $U$ one-tailed test, $p$ value <0.05 in both p2 and p15 comparisons; Supplementary Fig. 13). In summary, the clonal double kill-switches developed here are capable of stably blocking *Mycoplasma* growth even after 15 rounds of prior passaging.

**Table 1 List of the significant variants found after the IPTG-induction of KS1 in strain C5 at passage three (sample p3IPTG).**

| POS | QUAL | TOT | REFN | ALTN | FRAC | REF | ALT | MUT | AFF | EFF | IMPACT | INFO |
|---|---|---|---|---|---|---|---|---|---|---|---|---|
| 566053 | 1.95E-13 | 559 | 446 | 113 | 25.3 | CGCAAA | TGCTAT | LR208* | LacI4 | stop gained | HIGH | Expression of Nter fragment of LacI; Intact DNA-binding domain |
| 569125 | 0.00E+00 | 534 | 463 | 71 | 15.3 | ATCAAACT | GTCAAACC | K896fs | cas9B | frameshift variant | HIGH | Expression of truncated Cas9; Deletion of RuvCIII domain (affecting the Nuc lobe) and PAM-Interacting (PI) domain. The deletions of PI domain abolish cleavage activity |
| 566053 | 0.00E+00 | 271 | 224 | 47 | 21.0 | CGCAA | CACAT | LR208MC | LacI4 | missense variant | MODERATE | Mutation of a critical residue in the LacI protein for IPTG binding. It does not affect LacI structure |
| 566419 | 1.17E-13 | 609 | 571 | 38 | 6.7 | G | A | A87V | LacI4 | missense variant | MODERATE | Mutation that affects the hydrophobic surface of the IPTG binding pocket in the LacI protein |
| 566416 | 3.47E-14 | 281 | 255 | 26 | 10.2 | GGTGCGT | CGTTCTT | HAP86QER | LacI4 | missense variant | MODERATE | Mutation that affects the hydrophobic surface of the IPTG binding pocket in the LacI protein |
| 569127 | 8.08E-15 | 235 | 210 | 25 | 11.9 | CAAACT | CAACT | P897fs | cas9B | frameshift variant | HIGH | Expression of truncated Cas9; Deletion of RuvCIII domain (affecting the Nuc lobe) and PAM-Interacting (PI) domain. The deletions of PI domain abolish Cas9 cleavage activity |
| 571681 | 0.00E+00 | 481 | 469 | 12 | 2.6 | ATTTTTTTTGATA | ATTTTTTTGATA | N46fs | cas9B | frameshift variant | HIGH | Loss of reading frame that prevents expression of Cas9; The mutation affects the protein from its first domain RuvCI |
| 566420 | 0.00E+00 | 607 | 596 | 11 | 1.8 | C | A | A87S | LacI4 | missense variant | MODERATE | Mutation that affects the hydrophobic surface of the IPTG binding pocket in the LacI protein |
| 566423 | 9.10E-15 | 616 | 606 | 10 | 1.7 | G | A | H86Y | LacI4 | missense variant | MODERATE | Mutation that affects the hydrophobic surface of the IPTG binding pocket in the LacI protein |
| 571681 | 7.90E-15 | 191 | 182 | 9 | 4.9 | ATTTTTTTTGATACT | ATTTCTTTAGAAAACA | SIKK42* | cas9B | stop gained | HIGH | Expression of a small Nter fragment of the Cas9 that has only a portion of the RuvCI domain |
| 566193 | 7.10E-15 | 291 | 284 | 7 | 2.5 | CACGTCGAGAAAT | TATGACGAAAAAA | LFLDV158FFFVI | LacI4 | missense variant | MODERATE | Mutation of a critical residue in the LacI protein for IPTG binding. It does not affect LacI structure |
| 566423 | 1.70E-14 | 292 | 285 | 7 | 2.5 | G | A | H86Y | LacI4 | missense variant | MODERATE | Mutation that affects the hydrophobic surface of the IPTG binding pocket in the LacI protein |
| 568288 | 0.00E+00 | 231 | 225 | 6 | 2.7 | ATTTTTTTCAAAGGA | ATTTTTTCTATGGT | SF1173TI | cas9B | missense variant | MODERATE | Mutation in the PAM-recognition domain of the Cas9 protein; Deletion of the PI domain abolish Cas9 cleavage activity |
| 565983 | 2.26E-16 | 299 | 293 | 6 | 2.0 | C | A | W232C | LacI4 | missense variant | MODERATE | Mutation that affects the hydrophobic surface of the IPTG binding pocket in the LacI protein |
| 569609 | 0.00E+00 | 206 | 201 | 5 | 2.5 | ATACCTTTTTAATAG | TTACTTTTTTAATAG | GI736SN | cas9B | missense variant | MODERATE | Mutation in the mid helix A in RuvCII domain on Nuc lobe of the Cas9 that may affect the interaction with gRNA |
| 569662 / 565929 | 1.25E-15 / 6.21E-15 | 207 / 232 | 202 / 227 | 5 / 5 | 2.5 / 2.2 | ACTATCG / AACAATCCCCTCATTTAA | TCTTTCG / AACAATCCCCTTTTTTTA | DS718ER / LNE245* | cas9B / LacI4 | missense variant / stop gained | MODERATE / HIGH | Mutation in the RuvCII domain of Cas9 / Expression of Nter fragment of LacI; Intact DNA-binding domain |
| 570703 / 566136 | 0.00E+00 / 0.00E+00 | 237 / 243 | 232 / 238 | 5 / 5 | 2.2 / 2.1 | AAA / TAAGCGGGTCCCATCTTCGT | GAT / TTAGCCGGTCCCATCTTCGT | F372I / RL180* | cas9B / LacI4 | missense variant / stop gained | MODERATE / HIGH | Mutation in the domain REC1 of Cas9 / Expression of Nter fragment of LacI; Intact DNA-binding domain |
| 571104 / 565908 | 0.00E+00 / 0.00E+00 | 193 / 246 | 189 / 242 | 4 / 4 | 2.1 / 1.7 | CAAATAAGCCA / CGCCACGAG | CTATTAAGCCA / TGCCTCGTG | F238I / LV255HE | cas9B / LacI4 | missense variant / missense variant | MODERATE / MODERATE | Mutation in the domain REC2 of Cas9 / Mutation next to a Hotspot point mut Is phenotype; LacI incapable of induction |
| 568065 / 568703 | 0.00E+00 / 5.38E-15 | 200 / 200 | 197 / 197 | 3 / 3 | 1.5 / 1.5 | TATCTT / TAAAAG | CATCAT / CAATAC | EDN1250DDD / FFY1037LYC | cas9B / cas9B | missense variant / missense variant | MODERATE / MODERATE | Mutation in the PI domain of Cas9 / Mutation in the RuvCIII domain of Cas9 |

Columns correspond to genome loci (base pair position in the C5 genome; POS); Estimate of the probability of a polymorphism at the loci described by the record (QUAL). This value is presented in phred scale and takes into consideration both mapping and genotype quality for a specific variant; total number of reads obtained for the loci (TOT); Total number of reads matching the reference (REFN); number of reads presenting a variant (ALTN); Frequency of the variant normalised by the total reads found mapping that specific loci (FRAC); Reference sequence in the genome (REF); variant sequence (ALT); Degree of impact of the variant (IMPACT), which is classified as 'LOW' (synonymous mutations), 'MODERATE' (missense), 'HIGH' (nonsynonymous mutations, start or stop loss), or MODIFIER (variant found in an intergenic region); gene affected (AFF, in case the mutation is found intergenic, it is reported the closest downstream gene); Mutation or variant (MUT; an asterisk indicating stop codon); potential effect of the variant as given by snpeff[02] (EFF); and, in last column (INFO), short explanation of the expected effect of the variant in the mechanism of the kill-switch based on the literature (see references [48] and [81]). The full list of variants is provided in Supplementary Data 3.

These properties meet the challenging requirements of kill-switches and add to the utility of *M. pneumoniae* as a potential synthetic biology and vaccine chassis.

## Discussion

For the first time, there is a set of inducible systems for *Mycoplasma*. This work presented the adaptation to *M. pneumoniae* of well-characterised TF-repressor systems from gram-positive and gram-negative bacteria and phage. The expression of the different TFs does not appear to be toxic. More importantly, we showed it is possible to engineer synthetic inducible promoters for *Mycoplasma*. The promoter design is the essential step to make the inducible systems functional, especially considering that most heterologous inducible promoters did not meet the threshold in the test for *Mycoplasma* promoters. Altogether, the systems presented here represent the first gene regulation toolkit for *M. pneumoniae* and will lead the way to expand such tools for *Mycoplasma*. In the future, also, it will be interesting to explore the activity of TF-activators in *Mycoplasma*.

In this study, we used the gene regulation toolkit to engineer gene switches to limit *M. pneumoniae* growth. We presented two kill-switches, KS1 and TKS2, with different complexities but both based on the toxicity that an RNA-dependent nuclease produces in a cell incapable of repairing genomic DSB. The most challenging step in engineering such switches is to overcome the basal toxicity that would compromise cell viability. To this end, we optimised the OFF state of the kill-switch. This allowed us to achieve a good balance between undetectable basal expression and high toxicity once the kill-switch is induced. Remarkably, we successfully included a synthetic Theo-riboswitch in the design of the more complex circuit TKS2, despite the narrow dynamic range that this RNA elements provide. The inclusion of this RNA-regulatory factor created a synthetic Theo-addiction which can be exploited for *Mycoplasma* biocontainment.

This is the first time that circuit engineering has been applied in a minimal cell model. In this case, the genome-reduced bacterium *M. pneumoniae* exhibits a transcriptional regulation dominated by TF-independent mechanisms and with an important portion of unexplained variation associated with genetic stochasticity[23]. Therefore, it was a challenge to engineer orthogonal circuits in *Mycoplasma* based on canonical TF-repressors and riboswitches. We succeeded in engineering tight kill-switches based on CRISPR/Cas9 with reasonable long-term stability, despite the high mutation rate of *Mycoplasma* and their limited DNA repair machineries[7,79,83]. The durability of the biosafety circuits is a relevant concern for bioproduction[84]; engineered systems will always be subjected to the evolutionary pressure of the *Mycoplasma*, but, remarkably, we found that the kill-switches had been well silenced and do not increase that pressure, as the accumulation of mutations in the kill-switches is comparable to the rate found for the genome. Notably, we showed that a small inbuilt redundancy, like the duplication or combination of two kill-switches, is sufficient to decrease the escape frequency below the NIH recommended threshold for GMOs ($<10^{-8}$)[85]. This significantly increases the long-term stability of the biosafety circuits, which is predicted to reach up to 30 passages in the case of the double KS1 included in strain C20. This would be more than adequate for industrial bioproduction. In the future, it will be necessary to study the compatibility of the biosafety circuits with the expression of heterologous genetic elements included in the products.

The gene regulation toolkit for *Mycoplasma* opens a new horizon. Not only will it contribute to better tools to improve the understanding of *Mycoplasma* essential genes with unknown function(s), but, more importantly, together with the new genome engineering tools[39–43], it will qualify *Mycoplasma* as a true synthetic biology asset.

Furthermore, the *Mycoplasma* biosafety circuits show successful efficacy and robustness in vitro. In a follow-up, it will be useful to corroborate these results in the context of large-scale production, and more importantly, in vivo, where an immune system will have a role to play, too.

Recent advances in *Mycoplasma* research have led to designing different chassis with potential as live attenuated vectors, whether for vaccination or drug delivery systems[15,40,86,87]. For the aim of engineering live biotherapeutics for livestock or humans, it will be essential to ensure the biosafety of the products, during bioproduction as well. In this regard, we have presented two possible strategies to enhance the biocontainment of a MycoChassis: kill-switches that can be triggered upon demand and a synthetic addiction. Such genetic safeguards, which ensure confinement to controlled growth conditions[88–90], will contribute to gaining public confidence to bring the products to the market.

## Methods

**DNA cloning and construct construction**. Subcloning was performed using restriction enzyme sites, Gibson assembly[91], or reverse PCR and KLD reaction (NEB M0554). All constructs derive from an MTn vector, which could be either an MTn*par*[92] or MTn*cat*[93].

Designed promoter/operator pair sequences were evaluated in silico using the algorithm developed to predict promoters in the genome of *M. pneumoniae*[32]. The MTn-reporter platform used to test promoter candidates or RSs, is an MTn*par* derivative with a transcriptional terminator and a gene encoding a *mCherry* reporter[94] (Uniprot no. X5DSL3), with a 5′ end engineered with a short sequence coding for the leader peptide mp200[95] to facilitate translation (Supplementary Fig. 2A). We constructed two versions of the platform, one with the terminator TrrnMPN (A) and the other with the T142 (B). We inserted the promoter p438 and an RBS site upstream of the gene encoding the reporter to obtain the MTn used in the positive control of mCherry expression in *Mycoplasma*. Then, we generated a collection of MTn-reporter vectors changing the p438 sequence by the different promoter candidates using Gibson cloning. For the test of the RSs, the RBS region was substituted by the different RS sequences tested.

Genes coding for the AraR repressor (Uniprot no. P96711) and the AraE L-arabinose permease (Uniprot no. A0A6M3ZGR6) were isolated from *B. subtilis* str. 168 genomic DNA (ATCC 23857D-5). *Mycoplasma* codon-optimised genes purchased from GenScript for the optimisation of LacI (Uniprot no. P03023) and CI (Uniprot no. P03034) repressor proteins and, also, for the nucleases ApoRI (or AseI and SspI) and Cas9 (Uniprot no. Q99ZW2) used in the kill-switches. The genes coding for TetR repressor (Uniprot no. B1VCF0) and Cre recombinase (Uniprot no. P06956) proteins were extracted from the plasmid pΔMG_217Cre[45] kindly provided by Dr. J. Pinyol (IBB, UAB, Spain). Site-directed mutagenesis was used to generate the gene variants of *cas9* (B) and *cre* (B and C) genes. Promoters, ribosomal binding sites (RBSs), terminators and the gRNAs were ordered as primers (Sigma). PCR primers were also used to insert a short sequence coding for a leader peptide at the 5′ end of the genes or a 3′ end tag for protein detection.

The synthetic gene switches KS1 and derivatives were cloned into an MTn*par* vector. An MTn*cat* vector was used to clone the circuits TKS2 and KS1[3g], to generate strains C26 and C20, respectively.

All MTn constructs were confirmed by Sanger sequencing. All DNA sequences (primers, DNA parts, tools, and circuits) are listed in Supplementary Data 1B–F and GenBank files of the main plasmids are provided as supplementary data (all files are included in the supplementary file GenBank_MTns.zip).

**Bacterial strains and culture conditions**. Standard DNA cloning was performed with chemically competent cells TOP10 (Invitrogen), XL-1 Blue (Agilent), or DH5α F'1q (NEB). All constructs and circuits designed for *M. pneumoniae* were ensembled in an *E. coli* host. This step presents a bottleneck that impeded the generation of some constructs and their test in *M. pneumoniae*. To overcome this limitation, we exploited the use of codon TGA coding for W in *Mycoplasma* in relevant ORF sequences. We also used *E. coli* strains with constitutive expression of LacI or TetR to prevent unwanted expression from the synthetic IPTG- or ATc-inducible promoters included in some constructs.

All *M. pneumoniae* strains generated in this work are derived from M129 wild-type strain (ATCC 29342) and were cultivated in SP-4 medium, at 37 °C and 5% $CO_2$. Puromycin (3.3 μg/mL), chloramphenicol (20 μg/mL), L-arabinose (0.2% or 0,5%), anhydrotetracycline (ATc; 10 ng/mL or 50 ng/mL), IPTG (0.4 or 1 mM) or Theophylline (Theo; 0.5 mM) were added when appropriate. Polyclonal *M. pneumoniae* strains were generated by electroporation using 1 μg of MTn vectors as previously described[39]. Briefly, *M. pneumoniae* cells were grown in SP4 medium in

T25 (or T75) flasks at 37 °C and 5% $CO_2$ to late-exponential phase. Then cells were washed three times using electroporation buffer (272 mM Sucrose, 8 mM HEPES, pH 7.4) before being scrapped off in a final 1 mL volume of electroporation buffer and cell clumps disaggregated passing the cell suspension throw a 25 G needle ten times. For electroporation, 50 µL of the cell suspension was transferred to a 1 mm electroporation cuvette with 30 µL of electroporation buffer and 1 µg of the relevant MTn vector. The cells and DNA mix were kept on ice for 15 min before the electric shock at 1250 V, 25 µF and 100 ohms. Immediately after, the cuvette with the cells was transferred back on the ice for another 15 min before the cells were transferred into an Eppendorf with 500 µL fresh SP4 for a 2 h recovery time at 37 °C and 5% $CO_2$. Finally, a 1/100 dilution of the transformation was grown in SP4 flasks with the relevant antibiotics (and inducers) in T25 flasks at 37 °C and 5% $CO_2$ for a week when the transformed cells were recovered in 1 mL of SP4. Given the MTn vectors are randomly inserted in the mycoplasma genome, a pool of transformant cells with the pertinent MTn is recovered in SP-4 with selection antibiotic (puromycin when using a MTn*par* vector or chloramphenicol in the case of MTn*cat*) and theophylline when necessary. For clonal strains, after electroporation, individual clones were isolated in SP-4 plates with 0.8% agar and supplemented with the relevant antibiotic(s) and theophylline when pertinent. Specifically, *M. pneumoniae* clonal strains C5, C20 and C26 were isolated from SP-4 agar plates supplemented with either: puromycin; puromycin and chloramphenicol; or puromycin, chloramphenicol and theophylline. The location of the MTn in each clone was determined by A-PCR as previously described[39]. Briefly, genomic DNA from the clonal strain with an MTn unknown insertion is used in a low stringency PCR with the primers S78, with homology to a short fragment inside the MTn, and primer S79 which is a primer with a 5'unic sequence and the 3' randomised sequence NNNNNNGATTA ending in a highly repeated-sequence in the *M. pneumoniae* genome. Then, PCR products generated were purified as a pool and used as the template in a second PCR to isolate a single fragment with primers S78 and S80. This final fragment is sequenced to determine the insertion site of the MTn. The MTn location was then confirmed by PCR and sanger sequencing. For long-term studies, *M. pneumoniae* clonal strains C5, C20 and C26 were serially passaged every 48 h for 15th passages in SP-4 medium supplemented with puromycin, puromycin and chloramphenicol, or puromycin, chloramphenicol and theophylline, respectively.

All strains used in this work are listed in Supplementary Data 1G.

**Western blot.** For protein detection, total protein extracts were prepared from late-exponential phase cultures of *M. pneumoniae* strains for SDS-PAGE following standard procedures. Briefly, cells grown in 25 cm$^2$ flasks were washed three times with PBS and extracted with 50 µL of 1% SDS in TE prior to quantification using the Pierce BCA Protein Assay kit (Thermo Scientific). Ten or 25 µg of total protein extracts were loaded per sample in SDS-PAGE gels. Gels were transferred to nitrocellulose membranes using the iBlot system (Invitrogen). For immunodetection, membranes were blocked with 5% skim milk (Sigma) in PBS with 0.05% Tween 20 (PBST solution) overnight (o/n) at 4 °C and probed as previously described[96]. Briefly, membranes after blocking were incubated for 2 h at room temperature (RT) with the relevant primary antibody diluted in blocking solution (5% skim milk in PBST). Then membranes were washed three times with PBST and incubated for 1 h at room temperature (RT) with the relevant secondary antibody conjugated to horseradish peroxidase diluted in blocking solution. Finally, membranes were washed three times with PBST and Blots were developed with the Clarity™ Western ECL Substrate detection Kit (BioRad) and signals were detected in a ChemiDoc XRS + Gel Imaging System (BioRad). We used the following primary antibodies: mouse monoclonal 8C5.5 anti-mCherry (BioLegend 677702, 1:1000 dilution), mouse monoclonal M2 anti-Flag (Sigma F1804, 1:2000 dilution), mouse monoclonal V5-10 anti-V5 (Sigma V8012, 1:200 dilution), mouse monoclonal 0.T.81 anti-LacI (abcom 33832, 1:500 dilution), mouse monoclonal 9E10 anti-cMyc (Sigma M4439, 1:2000 dilution), rabbit polyclonal anti-TetR (Sigma T0951, 1:1000 dilution), mouse monoclonal 7.23 anti-Cre recombinase (BioLegend 900901, 1:500 dilution), and mouse monoclonal 7A9 anti-Cas9 (BioLegend 844301, 1:1000 dilution) antibodies. The secondary antibodies used were polyclonal anti-mouse IgG (Jackson Immune Research 515-035-003, 1:5000 dilution) or polyclonal anti-rabbit IgG (Jackson Immune Research 111-035-003, 1:5000 dilution) conjugated to horseradish peroxidase.

**Quantification of mCherry expression by fluorimetry.** Cell extracts were prepared from late-exponential phase cultures of *M. pneumoniae* strains grown in SP-4 medium (and relevant antibiotics) at different concentrations of the pertinent inducer in 25 cm$^2$ flasks. The adherent layer of cells was washed three times with PBS and then scraped off in 500 µL PBS. The cell suspension was passed ten times through a 25-gauge (G25) syringe needle to disaggregate cell clamps. The mCherry fluorescence intensity (Ex 585 nm/Em 625 nm) and Abs at 600 nm were measured from 150 µL of the different cell suspensions using a Tecan infinite 200Pro plate reader. Fluorescence intensity (arbitrary units) is normalised by Abs 600 nm showing induced expression of mCherry from three bio-replicates.

**Escape frequency determination by Colony forming units (CFUs).** *M. pneumoniae* strains were grown in 15 mL SP-4 medium with the relevant antibiotics, and Theo in the case of strain C26, in 75 cm$^2$ flasks. At the late-exponential phase, cells were harvested in a 1 mL fresh medium, and the cell suspension was passed ten times through a 25-gauge (G25) syringe needle to disaggregate cell clamps. Ten serial 1/10 dilutions of the cell suspensions were prepared and 10 µL drops of all cell dilutions were plated in triplicate. To determine the total CFUs in the cell suspensions, which defined the limit of detection of escapees, cells grew in SP4 plates with 0.8% agar, supplemented with antibiotic(s), and 0.5 mM Theo only in the case of the strain C26. To determine the escapees (cells with broken kill-switches), the cell dilutions prepared were plated in parallel in plates with the appropriate inducer(s) to trigger the kill-switches. Samples of the strains C5 and C20 were grown with 1 mM IPTG. Samples of the strain C26 were grown in plates without Theo and with 1 mM IPTG and 50 ng/mL ATc. Escape frequency was calculated as the ratio between CFUs found in plates with inducers and total CFUs.

**Time-course experiments.** *M. pneumoniae* strains were grown in a 5 mL SP-4 medium with the relevant antibiotics, and Theo when appropriate, in a 25 cm$^2$ flask. After 48 h, cells were harvested in 1 mL fresh medium, and the cell suspension was passed ten times through a 25-gauge (G25) syringe needle to disaggregate cell clamps. All cell suspensions from the different strains were adjusted to 0.1 of Abs 600 nm and later quantified by CFU counts.

*M. pneumoniae* growth was monitored using a Tecan Infinite 200Pro plate reader by determining the growth index value (ratio between the Abs430 nm and the Abs560 nm)[26]. A dilution of 1:100 or 1:1000 of the cell suspensions was used to inoculate 0.5 mL of fresh SP-4 with the appropriate antibiotics and inducers, and an aliquot of 150 µL of it was placed in a well in a 96-well plate with flat transparent bottom. Three independent bio-replicates were analysed. The experiments were performed at 37 °C collecting absorbance reads at 460 nm and 560 nm every 2 h for at least 5 days and up to 15 days. Importantly, in these long-term experiments, we observed that the volume of the samples in the wells is reduced due to evaporation affecting the values of the Abs560nm but not, significantly, the Abs430nm. Because of this effect, in long experiments, we included a sample with only SP-4 medium and no cells as a negative control of growth to define the baseline of the growth kinetics (black line in the graphs in Fig. 4).

For long-term experiments, *M. pneumoniae* growth was also analysed by genomic DNA. In this case, the cell suspensions of the different strains, prepared as explained above, were used to inoculate 15 mL of fresh SP-4 with the appropriate antibiotics and inducers. The 15 mL were then split in aliquots of 150 µL: one added in a 96-well plate for a growth index kinetics and the rest added in several wells in another 96-well plate for the genomic DNA time-course. Three bio-replicates were prepared for each condition. This second plate was incubated at 37 °C and, periodically, samples were harvested from one well for all three bio-replicates and genomic DNA was extracted using the GenElute™ Bacterial Genomic DNA Kit (Sigma). DNA was quantified by qPCR using primers S124 and S125 (Supplementary Data 1E). One µL DNA sample, 5 pmols of each primer and 5 µl LightCycler 480 SYBR Green I Master mix (Roche) and a Roche Applied Science LightCycler 480 Instrument (384-wells) were used (10 µL reactions). A serial 1:10 dilutions of a *M. pneumoniae* M129 genomic DNA sample, quantified by nanodrop, was used for the DNA standard curve.

The mCherry expression from synthetic promoters in *M. pneumoniae* was monitored by the fluorescence emission (Excitation 585 nm/Emission 625 nm) using a Tecan Infinite 200Pro plate reader. Samples were prepared following the same protocol as explained for the growth kinetics. Three independent bio-replicates were analysed. The experiments were performed at 37 °C collecting mCherry fluorescence emission and growth index reads every 2 h for 5 days. In these experiments, growth was monitored to ensure all samples behave similarly. Given the coloured SP-4 medium, the fluorescence reads have a high background signal at the beginning of the time-course, and it decreases with the acidification of the medium. In each experiment, a *M. pneumoniae* strain with an empty MTn was included to obtain the fluorescence background noise that is subtracted from the reads of the mCherry strains.

**Background correction and regression model of clonal strains data to predict growth curves over strain passages.** An effect of the medium volume reduction in the culture well is observed in the detection of the rate of absorbances Abs430/Abs560 for the clonal strains data types. Specifically, the volume reduction, due to evaporation, affects mainly Abs560 detection, while the effect on Abs430 is minimal. This is visible in SP4 data, consisting of the detection of Abs430/Abs560 in absence of the bacterium (Supplementary File 2). Linear regression was performed on SP4 data through the R (version 4.0.0) command *lm* for both Abs430 and Abs560, to capture the different entity of the effect of the volume reduction on the detection. Assuming both Abs430 and Abs560, in absence of the microorganism, should not change over time, the slope of the line obtained by linear regression was subtracted from the point-by-point mean of the original data, provided in triplicates. Slopes of the lines correspond to values of $1^{-08}$ for Abs430 and $-2^{-07}$ for Abs560. The rates of absorbances CAbs430/CAbs560 for SP4 were recomputed with the background-corrected absorbances and standard deviations were recalculated.

The difference between Abs430/Abs560 and CAbs430/CAbs560 was used to correct the data for all the strain types, for each condition and time point. All the

background-corrected curves, compared to the original data, are provided in Supplementary File 2.

CAbs430/CAbs560 data on clonal strains were used for building regression models to predict curve's parameters evolution over strain passages (Supplementary Fig. 15). The R package *Growthcurver*[97] (version 0.3.1) was used to extract $N_0$ (population size at the beginning of the exponential growth), $K$ (carrying capacity) and $r$ (growth rate) parameters for each strain passage. The population growth $N_t$ over time $t$ for each strain passage is described by Eq. (1):

$$N_t = \frac{K}{1 + \left(\frac{K - N_0}{N_0}\right)e^{-rt}} \tag{1}$$

The death rate was estimated through linear regression with the R command *lm*.

**Simulation of gene disruption in *M. pneumoniae* through metabolic modelling**. The most updated metabolic model of *M. pneumoniae* (iEG158_mpn)[82] comprises all genes involved in the metabolism of this microorganism that have been annotated. Each gene included in the model has been disrupted in silico by allowing no flux through the reactions linked to the specific gene. For each disrupted gene, growth has been simulated through Flux Balance Analysis[98], with biomass yield i.e. growth as an objective function, using *cobrapy* v. 0.5.11[99] in Python v. 3.4.4. Growth-after-disruption of the single genes has been compared to the wild-type growth of *M. pneumoniae* strain M129 in iEG158_mpn and percentages of decrease in growth have been computed (Supplementary Data 4).

**Whole-genome sequencing and variant calling analysis**. Genomic DNA samples were prepared from late-exponential phase cultures of *M. pneumoniae* strains C5, C20 and C26 grown in SP-4 medium (and relevant antibiotics and inducers). Cells grown in 75 cm² flasks were washed three times with PBS and followed by genomic extraction using the GenElute™ Bacterial Genomic DNA Kit (Sigma).

The London Biofoundry (Imperial College London) performed sequencing: Extracted genomes were prepared for sequencing with the Illumina Nextera DNA Flex kit (20018704) according to manufacturer recommendations via an automated implementation on an Opentrons OT-2 liquid handler. The resulting libraries were analysed and verified using a Fragment Analyzer (Agilent), Tapestation (Agilent) and Qubit (Thermo Fisher). Three libraries were pooled and diluted to 100 pM, and 20 µl loaded for sequencing and onboard de-multiplexing on an iSeq100 sequencer (Illumina).

A list of single nucleotide polymorphisms (SNPs), and insertions and deletions (INDELs) present in each sequenced population was obtained by paired-end mapping of raw sequenced reads against BWA[100]. Then, alignments obtained were analysed by freebayes[101] with relaxed stringency settings (minimum number of reads to trust a variation = 2, minimum total reads covering a site = 5, minimum fraction between alternative and reference reads = 0.001) and filtering out ambiguously mapped, low mapping and/or low base quality reads. This step also calculates the fraction of alternative reads (i.e., sequencing reads reporting a variant from the reference) with the total reads covering the same position, which can be used to evaluate the fixation of a variant in the population. Finally, we annotated the effect of each variant by using SnpEff[102,103] (Supplementary Tables S1, S2). To evaluate the functional impact of each variant found along samples or passages, we used the categorical impact annotated by SnpEff[102] that discriminates between 'low', 'moderate' and 'high' variants, corresponding to synonymous, missense, and frameshift/stop gained nonsynonymous mutations, respectively. A last impact category 'modifier' refers to variants found in intergenic regions. To evaluate the frequency at which mutations were found, variants were explored in terms of the number of variants normalized by the length in base pairs of the genomic region being explored. The regions considered included the cassettes C5 (6615 bp) and C20 (13,211 bp), and positions in the genome differentiated by those part of a gene (715,100 bp) or intergenic (101,294 bp).

**Biological materials availability**. DNA vectors with the tools or circuits (all vector maps are provided as GenBank files in the supplementary SupplMaterials.zip file), as well as, *M. pneumoniae* strains used in this study will be made available upon request.

**Reporting summary**. Further information on research design is available in the Nature Research Reporting Summary linked to this article.

## Data availability
The sequencing data generated in this study have been deposited in the ArrayExpress database at EMBL-EBI, under accession number E-MTAB-10981. The source and processed data for all fluorimetries, time courses, EF and qPCR generated in this study are provided with this paper as Supplementary files. Source data are provided with this paper.

## Code availability
The pipeline for whole-genome sequencing and variant calling analysis, references and processing steps required to generate the intermediate and supplementary files to replicate the variant calling analysis can be found at https://github.com/SMV818VMS/killswitch.

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

## Acknowledgements

We thank Marko Storch, Miles Priestman and Marta Ciechonska from the London Biofoundry for sequencing support. This work was funded by the European Union's Horizon 2020 research and innovation programme under grant agreement no. 634942 (MycoSynVac).

## Author contributions

A.B. and M.I. conceived the project and wrote the manuscript; A.B. performed all the experiments; A.B and S.M-V. analysed the iSeq data; E.G and V.A.M.d.S. performed the computational and modelling part. All authors reviewed and approved the manuscript. M.I. supervised.

## Competing interests

A.B and M.I declare that have filed a patent on this technology, as PCT Application Number PCT/GB2021/050184 (International Publication Number WO 2021/152301 A1, 5 August 2021). E.G. is currently employed as Project Officer at the European & Developing Countries Clinical Trials Partnership (EDCTP). E.G. declares no Horizon 2020 funding distributed by EDCTP has been deployed in this project and that the object of this manuscript is not involved in EDCTP scopes or funded grants. The remaining authors declare no competing interests.
