## [Peer Review File · Nature Communications]

Reviewers' Comments:

Reviewer #1:

Remarks to the Author:

Comments on:

Manuscript ID NCOMMS-21-38987

Title: A genetic toolkit and gene switches to limit Mycoplasma growth for a synthetic vaccine chassis

Author(s): Professor Isalan and colleagues

This study optimized existing transcriptional regulation system AraR, LacI, CI and TetR in Mycoplasma and combined them with Cas9 cutting to make kill-switch. The controllable kill-switch could be useful to develop attenuate vaccine for Mycoplasma pathogens.

The key goal of this research is to achieve the tight control of Mycoplasma gene expression. It involves codon optimization, promoter, RBS and cas9 design. This is an interesting research to the field of Mycoplasma as it lacks of gene regulatory tools.

The authors did a lot of things but logic flow is unclear. It should be improved.

Main concerns:

1. Figure 4A, the baseline shows that there are a few escape growth of strain C5, C20 and C26. But 4B only displays strain C5. How about C20 and C26? This escape frequency is higher than $1E-8$, contradictory to the claim in the page8 about the NIH threshold. Would cell death be evaluated using live/dead staining and FACS counting?
2. Figure 1B, it seems some data are below zero, please explain it or replot this figure. Figure 1C is one point of mCherry/A600, please indicate which point was used.
3. Kill-switch KS1 and TKS2 were introduced to Mycoplasma via electroporation. How the transformants were selected? It is helpful to describe it in the methods section. Since transposon was randomly inserted into the chromosome, would the insertion sites have any impact to the cell growth and function.

Reviewer #2:

Remarks to the Author:

The manuscript by Broto et al. describes the design and construction of a genetic toolkit and two kill-switches to limit Mycoplasma growth, for biosafety containment applications. Specifically, the authors tested four heterologous inducible promoter systems in *M. pneumoniae*, AraR, LacI, CI and TetR, and found that they are all functional. The authors engineered two gene switches, KS1 and TKS2, based on CRISPR/Cas9, to limit *M. pneumoniae* growth. The authors determined mutations in escape cells that contain the single kill-switch after several passages. They also demonstrated that combination of the two switches could provide efficient growth control with the addition of inducers. Finally, they demonstrated that the dual kill-switches continued to block cell growth even after 15 passages.

The manuscript is well written and the experiments were carried out very well. I recommend publication of this manuscript with minor revisions (see below).

1. The authors mentioned that single KS1 reduced cell growth but was inefficient, whereas KS1 TKS2 together had a strong kill effect. How about TKS2 alone? In Figure 3, TKS2 can control TetR, CL and Cas9 expression. Does it alone provide good cell growth limitation? It would be nice to see this data.
2. The authors described Mycoplasma will be used as a drug delivery vector. Did they test whether genome intergration of KS1 and TKS2 kill-switches interfered with foreign gene expression?
3. I was confused by the schematics of the circuits. In my view, it appears in these schematics that the inducing molecules inhibit the expression of the tested genes. Please clarify the schematics such that they show that the inducers enable expression.
4. In figure 3D, the authors should explain what "(2-)" means in the 2xTetR section.
5. In the Suppl. Fig. 4, the authors should explain in the legend that mCherry expression in these

stains is controlled by constitutive promoter p438. And that is the reason that the addition of inducer does not change mCherry level. This figure may be helped by showing a schematic of the mCherry expression cassette.

REVIEWER COMMENTS

Comments on:

Manuscript ID NCOMMS-21-38987

Title: A genetic toolkit and gene switches to limit *Mycoplasma* growth for a synthetic vaccine chassis

Author(s): Professor Isalan and colleagues

Reviewer #1 (Remarks to the Author):

This study optimized existing transcriptional regulation system AraR, LacI, CI and TetR in *Mycoplasma* and combined them with Cas9 cutting to make kill-switch. The controllable kill-switch could be useful to develop attenuate vaccine for *Mycoplasma* pathogens.

The key goal of this research is to achieve the tight control of *Mycoplasma* gene expression. It involves codon optimization, promoter, RBS and cas9 design. This is an interesting research to the field of *Mycoplasma* as it lacks of gene regulatory tools.

The authors did a lot of things but logic flow is unclear. It should be improved.

We thank the reviewer for the praise and the suggestion of improving the flow. As to the logic flow, our manuscript first describes how we built and tested different tools for the development of gene circuits; we then engineered the circuits and performed tests with the aim of improving them; finally, we studied their long-term stability. We understand the logic flow might appear unclear due to the numerous tools and circuits, so we have now described this workflow more explicitly in the text, specifically in the last paragraph of the introduction section.

Main concerns:

1. Figure 4A, the baseline shows that there are a few escape growth of strain C5, C20 and C26. But 4B only displays strain C5. How about C20 and C26? This escape frequency is higher than $1E-8$, contradictory to the claim in the page8 about the NIH threshold. Would cell death be evaluated using live/dead staining and FACS counting?

We thank the reviewer for this question. We studied the long-term stability of a kill-switch in the context of a minimal cell with the strain C5. We analysed growth, escape frequency (EF) and the fixation of inactivating mutations in the kill-switch at several strain passages to understand better the limitations that such a circuit has in *Mycoplasma*. The results indicate that the kill-switch was successfully silenced in *Mycoplasma*, and it evolves at the same rate as the cell where it is inserted. Given these results, we proposed to increase the half-life of such systems by implementing redundancy, for example, inserting a second kill-switch in the cells, generating strains C20 and C26. We compared the results obtained with these two new strains with a small redundant kill-system to the original C5 strain with a single kill-switch. The time-course experiments (Fig. 4A) and the iSeq analysis showed already the improvement of the system, so we did not find it necessary to do EF as well for strains C20 and C26 at late passages. We did analyse EF at an early passage and for these two strains and we can state that it is below the detection limit, as explained in the text (see the last two paragraphs for section III in results), because no escape colonies grew: therefore we were not able to display C20 and C26 in 4B.

It would be interesting to be able to count live/dead cells and we thank the reviewer for the suggestion. Unfortunately, it is not yet technically possible to do live/dead staining and FACS counting with *Mycoplasma* given their small size (less than 1 μm long).

2. Figure 1B, it seems some data are below zero, please explain it or replot this figure. Figure 1C is one point of mCherry/A600, please indicate which point was used.

The data shown in Fig. 1B correspond to the mCherry fluorescence of the strains, evaluated over time, after the subtraction of the background signal, that is obtained with an mCherry-defective strain. This is explained in the methods section "Time-course experiments" where we stated: "*In these experiments, growth was monitored to ensure all samples behave similarly. Given the coloured SP-4 medium, the fluorescence reads have a high background signal at the beginning of the time-course, and it decreases with the acidification of the medium. In each experiment, a M. pneumoniae strain with an empty MTn was included to obtain the fluorescence background noise that is subtracted from the reads of the mCherry strains.*" We now add a clarification in the Figure 1B legend: "(fluorescence in arbitrary units; background subtracted for autofluorescence - see *Methods*)"

As correctly pointed out by the reviewer, in Fig. 1C we show only 1 time point. We modified the figure legend as suggested by the reviewer: "(two days after induction)".

3. Kill-switch KS1 and TKS2 were introduced to Mycoplasma via electroporation. How the transformants were selected? It is helpful to describe it in the methods section. Since transposon was randomly inserted into the chromosome, would the insertion sites have any impact to the cell growth and function.

We thank the reviewer for the suggestion. As correctly pointed out, we do electroporation to introduce the MTn vectors to Mycoplasma. The transformants are selected using the antibiotics puromycin or chloramphenicol, depending on the MTn vector used in the transformation: MTnpar or MTnecat, respectively. Also, when the construct has a riboswitch, we included theophylline. We have now improved this explanation in the Methods section “DNA cloning and construct construction” and “Bacterial strains and culture conditions”, as the reviewer suggested.

Regarding insertion sites, we mainly studied the performance of the new tools and circuits in pools of cells (polyclonal strains) to diminish the impact that a particular insertion site of the MTn in the genome could have in the growth and function. Still, as is widely reported in the literature, a polyclonal strain generated by transposition on average has a reduced growth compared to the WT strain and, because of this, the strains used as negative controls in our experiments are also pools of cells carrying an empty MTn.

In this work, we also studied few clonal strains (selected because the colonies grew well) generated by transposition and isolated from agar plates. Then, as we explained in Methods, we determined the insertion site of the MTns by using the A-PCR method and sequencing. We found that the insertion sites of the MTns with the kill-switches, used to generate the clonal strains C5, C20 and C26, affected genes defined as non-essential in previous studies, or intergenic regions. As for the non-essential genes disrupted, mpn462 is annotated as *hypothetical protein* and no growth effect is expected. In the other case, gene mpn062 is annotated as a purine nucleoside phosphorylase and could have an impact in growth. Therefore, we simulated *in silico* the disruption of the genes comprised in the most updated metabolic model of *M. pneumoniae*. In the case of mpn062 the impact of disruption is 1.2% in growth (all the results of this simulation are included in Suppl. Table 4). The insertion site of the MTns can have an impact on the function of the circuits but, in the particular cases studied, those circuits were optimised in polyclonal strains and, when studied in clonal strains, no defects were observed. On the contrary, the clonal strains always had better performance.

In summary, we have studied clonal strains generated by transposition as we were limited by the engineering tools available at the time. In the future, brand new genome engineering tools for Mycoplasma will likely allow the design of the insertion site for the circuits to avoid growth or functional defects, as the reviewer points out.

Reviewer #2 (Remarks to the Author):

The manuscript by Broto et al. describes the design and construction of a genetic toolkit and two kill-switches to limit Mycoplasma growth, for biosafety containment applications. Specifically, the authors tested four heterologous inducible promoter systems in *M. pneumoniae*, AraR, LacI, CI and TetR, and found that they are all functional. The authors engineered two gene switches, KS1 and TKS2, based on CRISPR/Cas9, to limit *M. pneumoniae* growth. The authors determined mutations in escape cells that contain the single kill-switch after several passages. They also demonstrated that combination of the two switches could provide efficient growth control with the addition of inducers. Finally, they demonstrated that the dual kill-switches continued to block cell growth even after 15 passages.

The manuscript is well written and the experiments were carried out very well. I recommend publication of this manuscript with minor revisions (see below).

We thank the reviewer for the praise and the suggestions.

1. The authors mentioned that single KS1 reduced cell growth but was inefficient, whereas KS1 TKS2 together had a strong kill effect. How about TKS2 alone? In Figure 3, TKS2 can control TetR, CL and Cas9 expression. Does it alone provide good cell growth limitation? It would be nice to see this data.

We confirm the circuit TKS2 alone was analysed in the context of a WT cell and showed good control of TetR, CI and Cas9 (Fig. 3). This circuit TKS2 is turned into a kill-switch only when the recipient cell has a gRNA targeting the genome. To test the killing effect of TKS2, we inserted the circuit in the strain C5 with constitutive expression of a gRNA targeting the genome. This way, we showed the effect of the combined KS1+TKS2 circuits. Nevertheless, in the Suppl. Fig. 11B, we show the growth time course results of the polyclonal strain resulting from the insertion of the TKS2 into the C5 strain. In this case, we also tested the activation of only TKS2 when theophylline is removed, or ATc is added in the medium. The activation of only TKS2, with the addition of ATc, produces an important growth impairment, similarly to the results observed in the polyclonal KS1 strain (Fig. 2). As expected, growth impairment is stronger when the two kill-switches KS1+TKS2 are activated simultaneously.

2. The authors described Mycoplasma will be used as a drug delivery vector. Did they test whether genome integration of KS1 and TKS2 kill-switches interfered with foreign gene expression?

We thank the reviewer for the excellent question. Although very interesting, we did not study the interference with foreign gene expression because it exceeds the scope of this work. We decided to focus the study on engineering systems to limit Mycoplasma growth. As we mention in the manuscript, we believe that the good performance of the biosafety systems will add to the utility of *M. pneumoniae* as a potential synthetic biology asset. Definitely, in the case that these systems are to be implemented into a drug delivery Mycochassis, or similar, it will be necessary to test their compatibility with the expression of heterologous genetic elements included in the system, e.g. expression of antigens or drug delivery system.

We added in the discussion: *“In the future, it will be necessary to study the compatibility of the biosafety circuits with the expression of heterologous genetic elements included in the products.”*

3. I was confused by the schematics of the circuits. In my view, it appears in these schematics that the inducing molecules inhibit the expression of the tested genes. Please clarify the schematics such that they show that the inducers enable expression.

To improve the understanding of the schematics of the circuits, we moved the inducer inhibition arrows to reflect their direct inhibition of their corresponding TF. Please note that the activations work by inhibiting an inhibitor. e.g. IPTG inhibits LacI repression function, relieving repression, allowing activation.

4. In figure 3D, the authors should explain what “(2-)” means in the 2xTetR section.

We thank the reviewer, this is well noted. We have modified the figure legend to explain the meaning of (2-), as suggested by the reviewer: *“after two passages in theophylline-free medium (2-)”*.

5. In the Suppl. Fig. 4, the authors should explain in the legend that mCherry expression in these stains is controlled by constitutive promoter p438. And that is the reason that the addition of inducer does not change mCherry level. This figure may be helped by showing a schematic of the mCherry expression cassette.

We have modified the figure accordingly to the reviewer’s suggestions. We added a reference to the mCherry expression cassette from the Suppl. Fig. 2 in the figured legend: *“The expression of the different repressors did not affect the mCherry expression, as expected when it is driven by a constitutive promoter, like the p438 used in these analysis (See suppl. Fig. 2A for the schematic of the mCherry expression cassette).”*

Reviewers' Comments:

Reviewer #1:

Remarks to the Author:

The revised manuscript has addressed all of my questions. I recommend the publication.

Reviewer #2:

Remarks to the Author:

The revisions are acceptable.